# Imaging metal-like monoclinic phase stabilized by surface coordination effect in vanadium dioxide nanobeam

Zejun Li[1,*], Jiajing Wu[1,*], Zhenpeng Hu[2], Yue Lin[1], Qi Chen[3], Yuqiao Guo[1], Yuhua Liu[1], Yingcheng Zhao[1], Jing Peng[1], Wangsheng Chu[4], Changzheng Wu[1] & Yi Xie[1]

In correlated systems, intermediate states usually appear transiently across phase transitions even at the femtosecond scale. It therefore remains an open question how to determine these intermediate states—a critical issue for understanding the origin of their correlated behaviour. Here we report a surface coordination route to successfully stabilize and directly image an intermediate state in the metal-insulator transition of vanadium dioxide. As a prototype metal-insulator transition material, we capture an unusual metal-like monoclinic phase at room temperature that has long been predicted. Coordinate bonding of L-ascorbic acid molecules with vanadium dioxide nanobeams induces charge-carrier density reorganization and stabilizes metallic monoclinic vanadium dioxide, unravelling orbital-selective Mott correlation for gap opening of the vanadium dioxide metal–insulator transition. Our study contributes to completing phase-evolution pathways in the metal-insulator transition process, and we anticipate that coordination chemistry may be a powerful tool for engineering properties of low-dimensional correlated solids.

[1] Hefei National Laboratory for Physical Sciences at the Microscale, CAS Center for Excellence in Nanoscience, and CAS Key Laboratory of Mechanical Behavior and Design of Materials, University of Science and Technology of China, Hefei, Anhui 230026, China. [2] School of Physics, Nankai University, Tianjin 300071, China. [3] i-Lab, Suzhou Institute of Nano-Tech and Nano-Bionics, Chinese Academy of Sciences, Suzhou, Jiangsu 215123, China. [4] National Synchrotron Radiation Laboratory, University of Science and Technology of China, Hefei, Anhui 230029, China. * These authors contributed equally to this work. Correspondence and requests for materials should be addressed to C.W. (email: czwu@ustc.edu.cn).

As a prototype correlated material, vanadium dioxide ($VO_2$) exhibits an abrupt first-order metal-insulator transition (MIT) near room temperature of $\sim 340$ K ($T_{MIT}$), accompanied by a lattice change on the picosecond timescale[1–4]. This metal-insulator transition in $VO_2$ raises hope for wide potential applications, ranging from ultrafast switching techniques, Mottronics to memristors[5–8]. In the metallic rutile phase above $T_{MIT}$, vanadium atoms are linear and equally spaced along the $c$ axis; on entering the insulating monoclinic (M1) phase below $T_{MIT}$, they are slightly distorted into dimerized zigzag chains[9–11]. Pioneering studies indicate that Mott correlation and Peierls distortion play key roles in triggering the MIT[5,12–14]. However, despite intensive theoretical and experimental efforts, the detailed changes of electronic and lattice structure for the $VO_2$ MIT process, including the sequence of the phase transition and the origin of the insulating gap, are still under debate.

Stabilization and precise identification of intermediate states hold promise for understanding the full physical nature of the MIT process. For example, as a transient intermediate state, the discovery of an insulating M2 phase, comprising one half of the undimerized V chains, helps us to rule out a pure Peierls mechanism in the gap opening of $VO_2$ MIT[8,15]. The observed correlated metallic rutile puddles, as an intermediate state, exhibited mass divergence with a signature of electronic correlation effect across the phase transition[16]. However, there is still a long way to go for a full understanding of the MIT in $VO_2$, especially for precisely identifying electronic intermediate states to complete phase-transition pathways. The great challenge comes from the limitation of currently available characterizations, in that it is indeed hard to capture intermediate electronic states in spatial isolation because of their transient occurrence. Progress has been made to monitor the evolutions of lattice and electronic structures near the $VO_2$ MIT process, relying upon indirect approaches such as Raman spectra, soft X-ray absorption spectroscopy, temperature-dependent resistance measurements, and the combination of ultrafast electron diffraction, time-resolved infrared transmittance and so on[17–20]. As an experimental fact, electronic and lattice structures of the intermediates were not easy to be directly taken out from the transient process; and it therefore remains an open question how to stabilize the intermediate state that is critical to unravel the physical picture in correlated systems.

Here we demonstrate a surface coordination route to experimentally stabilize and directly image an intermediate state in the metal-insulator transition (MIT). As a prototype MIT material of $VO_2$, we capture an unusual metal-like monoclinic $VO_2$ intermediate phase. A coordination effect of L-ascorbic acid (AA) on $VO_2$ nanobeams induces one-dimensional (1D) charge-carrier density reorganization and a nonequilibrium metal state in monoclinic $VO_2$. Metal-like monoclinic $VO_2$ stabilized over several micrometers exhibits higher charge-carrier density compared with the insulating counterpart, suggesting that the insulating gap of M1 is correlated with orbital-selective Mott correlation. Our findings may open an alternative coordination-chemistry direction in modulating low-dimensional correlated materials.

## Results

### Chemical treatment of $VO_2$ nanobeams in AA solution.
In our experiments, an exotic evolution of electronic phases was triggered along the $VO_2$ nanobeams by chemical treatment. Single crystal $VO_2$ nanobeams were grown according to our earlier report[21]. $VO_2$ nanobeams were then treated in AA aqueous solution at $80\,^\circ$C (see Methods for details). Fig. 1b shows the optical images of the pristine and AA-treated $VO_2$ nanobeams with increasing treated time, revealing a phase evolution at room temperature. The optical reflections of different $VO_2$ phases are remarkably distinct, in which bright and dark domains correspond to insulating M1 and metallic rutile phases, respectively[22]. We can see that the pristine $VO_2$ nanobeam exhibited bright reflection at room temperature. Strikingly, after AA treatment, dark domains initially emerged at the two ends of the $VO_2$ nanobeam, resulting in the coexistence of dark and bright domains. Then, the terminal dark domains continued to expand along the nanobeam with increasing treating time and eventually formed an entirely dark-colored nanobeam. These optical images revealed that the AA treatment induced stabilization of metallic rutile phase, emerging from two ends of the $VO_2$ nanobeam and gradually evolving along the nanobeam. To further verify the phase transition, selected area electron diffraction (SAED) was performed on a single AA-treated $VO_2$ nanobeam (Supplementary Fig. 1), which indeed confirmed that the terminal dark parts were indexed to rutile structure and the middle bright region still remained as M1 structure. Comparable experiments were also performed to understand the appearance of metallic rutile domain, $VO_2$ nanobeams were treated in pure water without adding AA, and other conditions were kept unchanged. As exhibited in Supplementary Fig. 2 and Supplementary Note 1, the $VO_2$ nanobeam exhibits no dark domains emerging at the two ends, and the Raman spectrum demonstrates M1 phase at room temperature after treatment in pure water, indicating that the emergence and stabilization of rutile metal phase are attributable to the AA treatment.

### Metal-like monoclinic phase in AA-treated $VO_2$ nanobeams.
More strikingly, we observed a metal-like monoclinic $VO_2$ metastable phase in the AA-treated $VO_2$ nanobeam with monoclinic lattice but metal-like infrared optical characteristic. The metastable phase has been directly discovered through a thermal treatment. Figure 2a shows a typical AA-treated $VO_2$ nanobeam with about $3.8\,\mu$m long dark rutile domain in the end, which is cantilevered from the edge of the silicon substrate to thoroughly wipe out the effect of substrate strain. As it warmed up to 350 K, the $VO_2$ nanobeam turned into homogeneous rutile phase (Fig. 2b). Of note, the terminal dark domain elongated to $6.9\,\mu$m after cooling down to room temperature (Fig. 2c). This would imply that a part of the bright domain adjacent to the boundary could easily convert into rutile phase through the thermal loop, which indicated a significantly typical thermo-metastable property. Herein, we nominated this part of domain to a metastable state. In order to determine the structure of the metastable phase, SAED was performed on the same spot of the optically changed region before and after heating. As is depicted in the inset of Fig. 2a, before heating, the encircled region exhibited M1 structure. After the thermal treatment, SAED of the same region was assigned to rutile structure (inset of Fig. 2c). In addition, the result of corresponding Raman spectra was consistent with that of the SAED patterns (see Supplementary Fig. 3). Thus, it suggested that this $VO_2$ metastable phase retained the monoclinic lattice feature in the AA-treated nanobeam.

Metal-like optical behaviour of this monoclinic metastable phase is gained from scattering scanning near-field infrared microscope (s-SNIM) with resolution of about 10 nm (see Supplementary Fig. 4 for details). The s-SNIM enables us to distinguish electronic phases according to their optical contrast, where metallic state shows enhanced scattering amplitude compared with the insulating counterpart[16]. The representative near-field image of AA-treated $VO_2$ nanobeam as a function of the position along the nanobeam at room temperature is

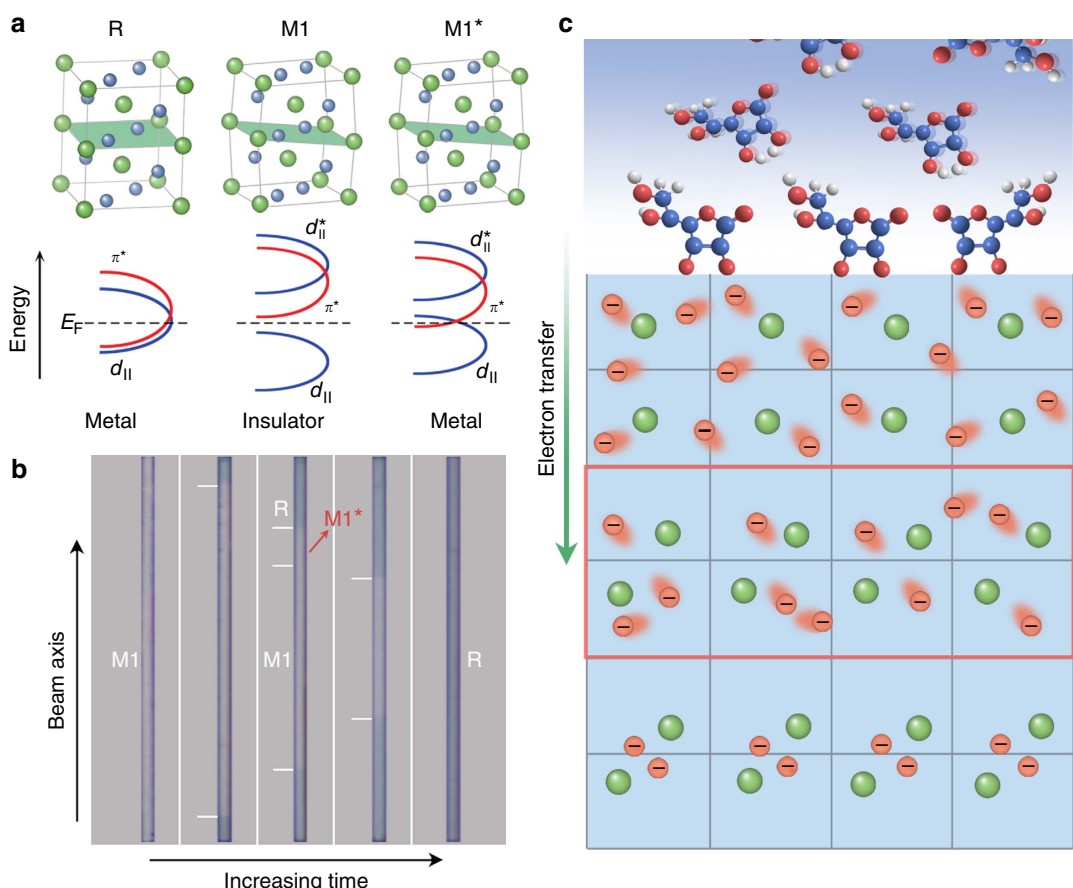

**Figure 1 | Phases of VO₂.** (**a**) Crystal structures of different VO$_2$ phases with the corresponding band structures. Left of panel: metal rutile phase, Middle of panel: insulating monoclinic (M1) phase, Right of panel: intermediates of metal-like monoclinic (M1*) phase with the proposed band diagram from ref. 20. (**b**) Room temperature optical images of the AA-treated VO$_2$ nanobeams with increasing treatment time, showing the phase evolution. The treatment time from left to right is: 0, 1, 3, 5 and 8 h. The stabilized M1* lies between R and the M1 host. (**c**) Schematic of the phase transition induced by AA treatment. The portion marked in the red rectangular frame is the M1* phase with high-density itinerant electrons.

displayed in Fig. 2d,e, showing three distinct scattering signals (indicated by white, red and yellow arrows). We can see that the middle region guided by the white arrow provides a low scattering signal, while strong amplitude of the scattering is mapped on the end of the nanobeam. This optical feature demonstrated the insulating state of the middle M1 phase and metal state in the end rutile part, matching well with the above optical reflection and SAED data.

In particular, the metastable domain guided by the red arrow showed a relatively enhanced scattering signal, which was about more than two times that of the middle insulating M1 phase. The remarkable increase of the infrared scattering indicated a metal-like electronic state in the metastable phase[16]. In contrast, we have also mapped near-field image of the same nanobeam after the heating treatment mentioned above. As sketched in Fig. 2f,g, we observed a clear increase of the scattering signal in the metastable metal-like electronic region to an equal intensity as that of the terminal rutile phase, which is consistent with the above results of optical contrast and SAED. Together with the SAED results, a metal-like monoclinic VO$_2$ phase has been stabilized and isolated in the AA-treated VO$_2$ nanobeams. To the best of our knowledge, this is the first experimental stabilization of metal-like monoclinic phase of VO$_2$, which allows us to investigate its precise lattice and electronic structure.

**High charge-carrier density in metal-like monoclinic domain.** We further investigated the charge-carrier behaviour of the

metal-like monoclinic VO$_2$ using dielectric force microscopy (DFM), a contactless imaging technique with nanometer-scaled spatial resolution[23,24] (Fig. 3a and Methods section). As is known, the dielectric response of a material is determined by its charge-carrier density and mobility. In the case of our VO$_2$ nanobeam, the carrier mobility of the insulating monoclinic phase and metallic rutile phase is similar[25], thus the dielectric response reflects its charge-carrier density. Figure 3b showed the topographical profile of an AA-treated VO$_2$ nanobeam from DFM, presenting a regular morphology. The single AA-treated VO$_2$ nanobeam has been measured under gate voltage ($V_g$) from −4 to +4 V by DFM, and the representative DFM image of fixed bias voltage at $V_g = -4$ V is shown in Fig. 3c. It is found that the dielectric response exhibited three different types of DFM signal strength. The dielectric signal in the end domain is much stronger than that of the middle, indicating higher carrier density in the end region, which agrees well with above results. The striped appearance was caused by the slight signal fluctuation and noise from DFM measurements as well as the slight thickness of our nanobeams. Of note, the dielectric signal in the metal-like monoclinic VO$_2$ domain (green frame area) was weaker than end area but clearly stronger than the middle, indicating that the stabilized metal-like monoclinic VO$_2$ possessed high charge-carrier density (Fig. 3d). The results are further supported by quantitative DFM signals at different $V_g$ from −4 to +4 V (Supplementary Fig. 5 and Supplementary Note 2), which convincingly suggest charge-carrier density in the stabilized

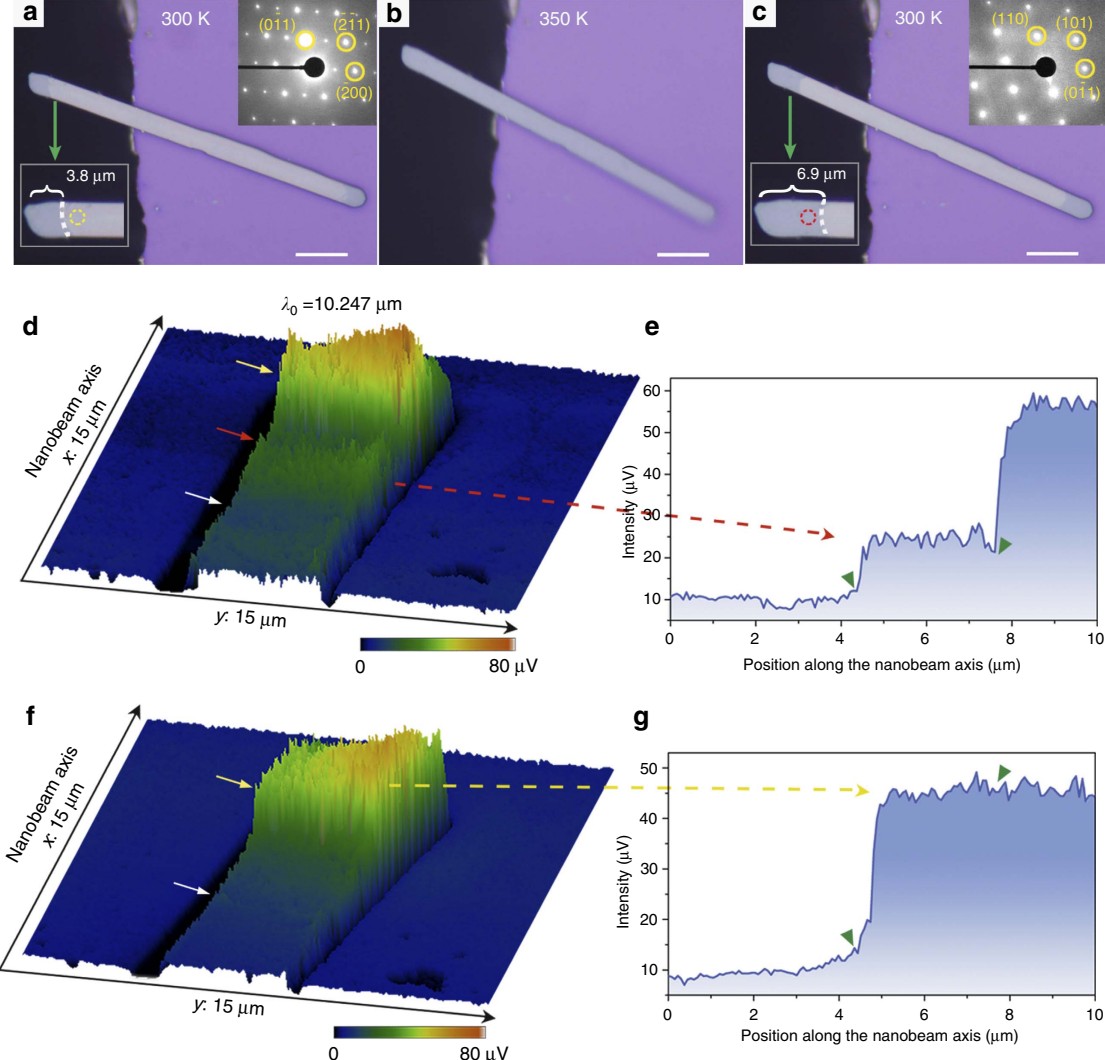

**Figure 2 | Metal-like monoclinic phase of VO₂.** (a–c) Optical images of an AA-treated VO$_2$ nanobeam obtained through heating treatment. The inset shows the corresponding SAED of the same region marked by the yellow and red circles before and after heating treatment, respectively. Scale bars, 10 μm. (d,f) Images of the near-field scattering amplitude for the AA-treated VO$_2$ nanobeam before and after heating treatment, obtained by the s-SNIM with the incident wavelength ($\lambda$) of 10.247 μm. (e,g) The corresponding near-field scattering amplitude as a function of the position along the nanobeam.

metal-like monoclinic VO$_2$ region is higher than the middle region, while weaker than the end region.

**Surface molecular coordination.** Coordinal bonding of AA species on the treated VO$_2$ nanobeam was revealed by ATR-FTIR spectra. As shown in Fig. 4a, the spectrum of AA-treated VO$_2$ sample resembles that of the pure AA molecules, thus confirming the connection of AA species to the VO$_2$ nanobeams. In contrast, the spectrum of pristine VO$_2$ sample exhibits no signal, further indicating the attachment of AA species on AA-treated VO$_2$ sample. Of note, the enolic C–O–H scissoring and stretching vibration of pure AA molecule at 1,271 and 1,318 cm$^{-1}$ disappeared in the AA-treated VO$_2$ spectrum[26]. Instead, a new peak occurred at wavenumber of 1,390 cm$^{-1}$, which indicated that the enolic C–O–H of AA molecule reacted with the VO$_2$ during the solution treatment and formed a new connecting bond with the surface atoms of VO$_2$. Such surface modification is reminiscent of the C–O–Ti bidentate complexes in which the enolic C–O–H of AA molecules binds to the solid surface[27,28]. In this regard, we ascribed that the new peak at 1,390 cm$^{-1}$ was stemming from the formation of analogous C–O–V complexes on

the AA-treated VO$_2$ surface (Fig. 4b), which was further supported by theoretical simulation (Supplementary Note 3). Notably, some other electron-donating molecules, such as oxalic acid, ethylene glycol, glucose and NaBH$_4$, have also been used to treat VO$_2$ nanobeams as comparable experiments. However, the corresponding phase evolution of VO$_2$ nanobeams has not been observed as that appearing in AA treatment (see Supplementary Fig. 6 and Supplementary Note 4 for details).

**Discussion**

Surface coordination of VO$_2$ nanobeam with AA molecules led to charge transfer at the AA/VO$_2$ interface. The AA molecule has a particular enolic structure, which exhibits strong electron donating and coordination ability. The surface chelation of AA molecules with VO$_2$ resulted in the interfacial p-d hybridization (V—3d orbits and O—2p orbits of AA species), and led to electron-transfer complex (Fig. 4c), which was supported by the calculations (Supplementary Fig. 7 and Supplementary Note 5). From the above results of the DFM, the striking feature of high charge-carrier density in metal-like monoclinic VO$_2$ domain also revealed the external electron injection process during the AA

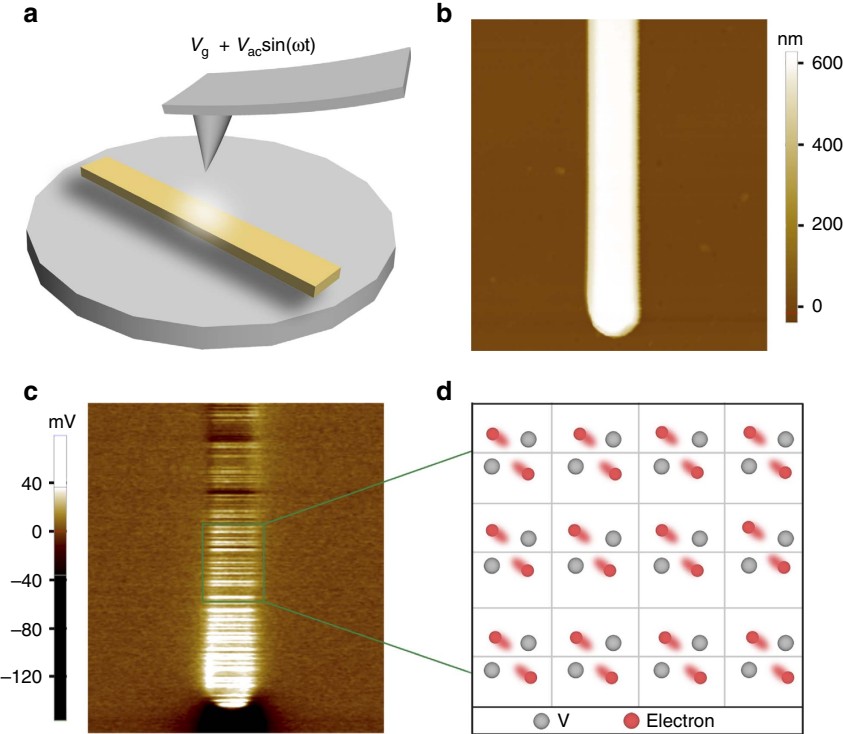

**Figure 3 | Electronic state of the metal-like monoclinic VO₂ at room temperature.** (**a**) Sketch of the experimental setup of DFM. (**b**) Topographical image of an AA-treated VO₂ nanobeam. (**c**) Dielectric response of AA-treated VO₂ nanobeam. (**d**) Schematic simulation of the metal-like monoclinic VO₂ with high charge-carrier density.

treatment, giving a convincing evidence of the electron transfer from the AA molecules to VO₂ nanobeam. In additon, optical images suggested that the electron injection was along V–V atomic chain from the nanobeam ends toward the middle. Thus the rutile phase domain grows in from the ends of the nanobeam (see Supplementary Discussion for details). In the dark end domains, the injected electrons were adequate to stabilize metallic rutile phase to room temperature. While in the metal-like monoclinic domains, the injected electrons were insufficient to suppress Peierls instability; thus, for the metal-like monoclinic phase, its lattice was distorted to monoclinic structure but a nonequilibrium metallic electronic state was induced.

Stabilization of metal-like monoclinic phase of VO₂ originated from the orbital-selective charge density reorganization. In VO₂ phases, the V ions have three $t_{2g}$ orbitals that is $d_{||}$ ($d_{xy}$) and $\pi^{\star}$ ($d_{xz}$, $d_{yz}$) orbitals. It has been suggested that the electron occupancy of the three $t_{2g}$ orbitals determines the electronic phase of VO₂ (refs 13,20,29). In rutile phase, these three $t_{2g}$ orbitals almost completely overlap at the Fermi level, exhibiting isotropic metal electronic state. When VO₂ transforms to insulating monoclinic phase, the V $d_{xy}$ orbital occupation increases and the orbital occupation of $d_{xz}$, $d_{yz}$ is reduced, resulting in the redistribution of charge density and forming a more 1D electronic state along the V atom chains. In our case, the chelated AA molecules transfer electrons to VO₂ (rutile) nanobeam. When the injected electrons are sufficient, the rutile metal state can be stabilized to room temperature. While in the metal-like monoclinic region, owing to less injected charges, the electron density in this portion is not sufficient enough to suppress the Peierls distortion to monoclinic lattice. However, the injected electrons would be preferentially accommodated in the $d_{xz}/d_{yz}$ subshells, leading to a 1D reorganization of charge-carrier density between the $d_{xy}$ and $d_{xz}/d_{yz}$ subshells. This increased carrier density in the $d_{xz}/d_{yz}$ orbitals could partially suppress the Mott correlation and induce a nonequilibrium metal electronic

state. Of note, the metal-like monoclinic phase was limited to a small section between the rutile phase domain and remaining M1 domain (see Supplementary Discussion for details). Thus, these results demonstrated that the interfacial electron injection induced 1D reorganization of charge-carrier density and stabilized the metal-like monoclinic VO₂ phase.

Metal-like monoclinic phase of VO₂ provides an insight into VO₂ MIT. Our study indicates a non-congruent structural and electronic transition in MIT of VO₂, and the MIT is the result of a collaborative Peierls and Mott transition. Importantly, the Mott correlation is closely associated with the formation of insulating M1 phase. The charge occupancy in the $t_{2g}$ orbitals influences the Mott instability in VO₂ MIT, which is favourable to the prediction of orbital-selective Mott transition[29]. In a word, competing orbital occupation of the charge plays a key role in determining the electronic phases and lattice in VO₂ system.

In summary, we successfully developed a surface coordination approach to stabilize and directly image a phase-transition intermediate in the MIT of VO₂ at room temperature. We experimentally stabilized and isolated an unusual metal-like monoclinic VO₂ phase in the MIT process. The AA molecules chelated on the surface of VO₂ nanobeams induced 1D charge-carrier density reorganization, triggering a nonequilibrium metallic state in monoclinic VO₂. Stabilization of metal-like monoclinic phase opens the doors for precisely identifying the phase-transition intermediates to understand the complete process of VO₂ MIT. Our results revealed the critical role of competing charge distribution in phase transitions of correlated systems. We hope that the coordination effect may provide a new perspective to regulate electronic properties of low-dimensional correlated materials.

## Methods
**AA solution treatment of VO₂ nanobeams.** Single crystalline VO₂ nanobeams were transferred from the grown substrate onto clean, thermally oxidized

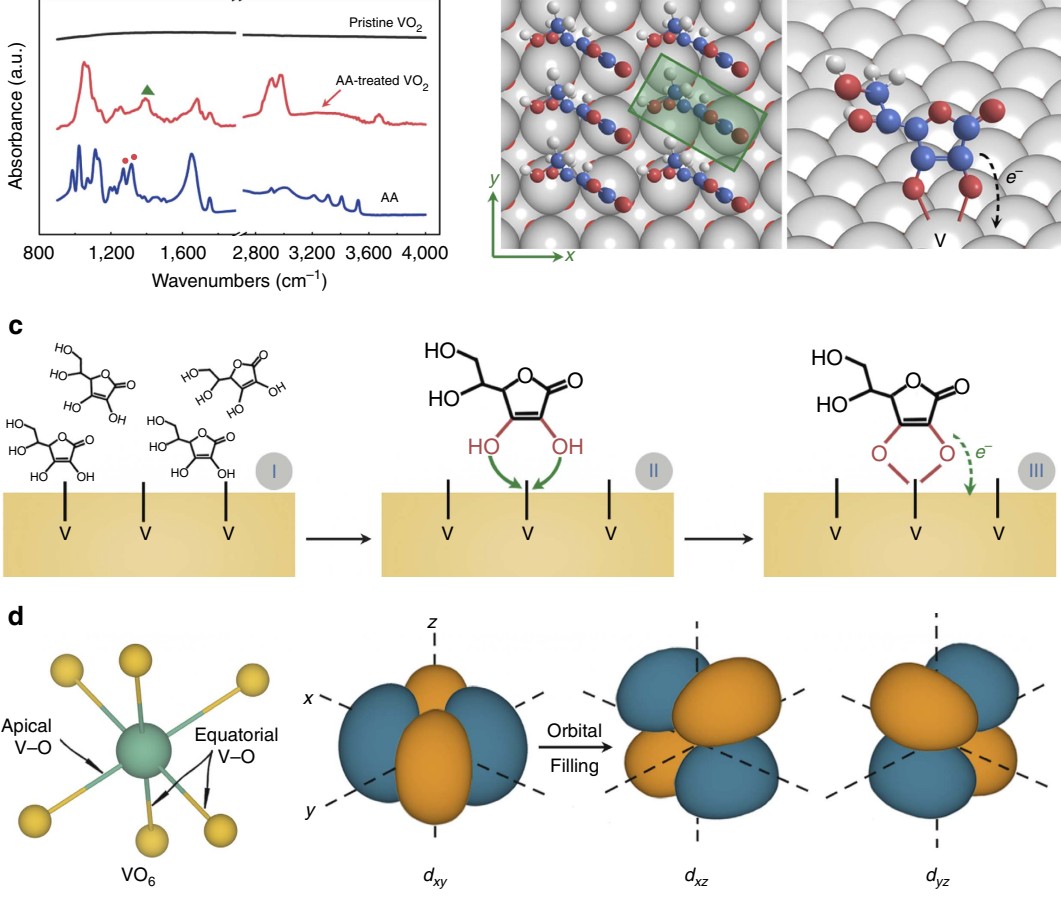

**Figure 4 | Mechanism of stabilization of metal-like monoclinic VO₂ by surface coordination in the nanobeam.** (**a**) Attenuated total reflectance Fourier transform infrared (ATR-FTIR) spectra of pristine VO₂, AA-treated VO₂ sample and pure AA molecules, respectively. The enolic C–O–H scissoring and stretching vibration of AA molecule at 1,271 and 1,318 cm$^{-1}$ labelled by red dots disappear in the AA-treated VO₂ spectrum, and a new peak labelled by green triangle occurs at 1,390 cm$^{-1}$ in the spectrum of AA-treated VO₂ sample. (**b**) Depiction of the AA molecules absorbed on the end face of VO₂ nanobeam with bidentate binding to surface vanadium atoms. (**c**) Diagram for the stepwise reaction of surface coordination of AA molecules on VO₂ nanobeam. (**d**) Schematic of vanadium $t_{2g}$ orbitals in the VO₂ phase.

silicon wafers. These substrates were put into a three-necked flask containing 80 ml 1 M AA aqueous solution, and the mixture was refluxed at 80 °C for several hours with nitrogen protection. The treatment time referred to the reflux reaction time, which ranged from 1 to 8 h. After cooling to room temperature, the samples were immersed in distilled water and ethanol several times to wash away the residual AA solution on the substrates, and then blow dried with nitrogen for further characterization and measurements.

**Transferring the AA-treated VO₂ nanobeams.** The AA-treated VO₂ nanobeam was transferred onto the edge of silicon substrate using silver needle under Olympus optical microscope. In a typical procedure, the silver needle slowly approaches and touches the AA-treated VO₂ nanobeam on the SiO₂/Si substrate. Then the AA-treated VO₂ nanobeam is adsorbed on the tip of the silver needle. Subsequently, the silver needle is moved slowly to the edge of another SiO₂/Si substrate and the nanobeam is put down on the edge of the substrate. Finally, slight adjustment made one end of the AA-treated VO₂ nanobeam suspend on the edge of the substrate.

**SAED measurements.** The optical image of AA-treated VO₂ nanobeam on the SiO₂/Si substrate was taken and the length of the dark rutile domains along the nanobeam was recorded. Then, the AA-treated VO₂ nanobeam was transferred onto the copper grid using a silver needle. During the SAED measurements, the boundary of dark and bright domains was first determined according to the optical image. Then the SAED for the metastable region adjacent to the boundary was performed and marked the corresponding position. Subsequently, the copper grid was taken out and heated on a hot plate to 350 K. After cooling to room temperature, the copper grid was put back and found the same region on the

nanobeam as before heating treatment for SAED. Finally, the SAED pattern of the same region was obtained.

**DFM measurements.** For the DFM measurements, single AA-treated VO₂ nanobeam with M-I-M domain pattern was transferred to a p$^{++}$-Si (100) substrate with 50 nm thermal oxide. An XE-120 microscope (Park Systems Corp., Suwon, Korea) placed in a N$^2$ filled glove box was used for all DFM measurements. Conducting AFM tips (NSC18/Cr-Au, Mikromasch, Tallinn, Estonia) with a resonance frequency of about 80 kHz and spring constant of about 2.8 N m$^{-1}$ were used as probes. The experimental setup of DFM was developed from a double-pass electrostatic force microscopy imaging process with slight modifications. In the first pass, the standard AC mode imaging was performed on the sample to acquire a topographic scan line, which was linearly fit to obtain the topographic baseline; in the second pass, the oscillation of the cantilever at resonance frequency was turned off, and the tip was lifted up to a certain height (typically 10–20 nm) and scanned in a trace parallel to the topographic baseline obtained in the first scan with a bias voltage $V = V_g + V_{ac}\sin(\omega t)$ applied between tip and Si substrate.

**Characterization.** The optical images were obtained using an Olympus microscope. SAED patterns were acquired using a JEOL JEM-ARF200F microscope. Raman spectra were detected on a LABRAM-HR confocal laser micro Raman spectrometer. The s-SNIM near-field images were performed on NeaSNOM (Neaspec GmbH Co.) A Shimadzu IRPrestige-21spectrometer with a ZnSe ATR cell (Specac Ltd., Woodstock, GA) and a ZnSe grating polarizer (PIKE Technologies Inc., Madison, WI) was used to collect ATR-FTIR spectra.

**Data availability.** The data that support the findings of this study are available from the corresponding author on reasonable request.

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

## Acknowledgements

We thank Dr Jinbo Cao at GE and Dr Wen Fan at USTC for valuable advice. This work was financially supported by the National Basic Research Program of China (2015CB932302), the National Natural Science Foundation of China (U1432133, 11321503, J1030412, 11404314), National Young Top-Notch Talent Support Program, the Fok Ying-Tong Education Foundation, China (Grant Nos. 141042, 151008), the China Postdoctoral Science Foundation (Grant No. 2016M600483) and the Fundamental Research Funds for the Central Universities (WK2060190027, WK2340000065), Anhui Provincial Natural Science Foundation (1708085MA06). We would like to thank Dr Lin-Jun Wang and Xiaolei Wen (USTC Center for Micro and Nanoscale Research and Fabrication) for discussions about the contact fabrication and s-SNIM measurements, and Dr Yuyan Han (High Magnetic Field Laboratory of Chinese Academy of Science) for assistance with the sample measurements.

## Author contributions

C.W. conceived the idea and experimentally realized the study, co-wrote the paper and supervised the entire project and is responsible for the infrastructure and project direction. Z.L. and J.W. contributed equally to this work; they experimentally realized the study, analysed the data and co-wrote the paper, and these authors were assisted by Y.L., Q.C., Y.G., Y.L., Y.Z., J.P. and W.C. The theoretical calculations were performed by Z.H. Y.X. supervises the whole procedures. All the authors discussed the results, commented on and revised the manuscript.

## Additional information

**Competing interests:** The authors declare no competing financial interests.

**Publisher's note**: 

