## [Peer Review File · Nature Communications]

Reviewers' comments:

Reviewer #1 (Remarks to the Author):

In this manuscript the authors present a variety of measurements on vanadium dioxide (VO₂) nanobeams that were subject to a treatment procedure involving L-Ascorbic acid (AA). The authors interpret these results as suggesting that the AA treatment stabilizes a monoclinic metallic phase of VO₂ at room temperature (similar to that previously observed in time-resolved studies of photoexcited VO₂, Ref.20 in the current manuscript) along a small section of the treated nanobeam located between insulating M1 and metallic R domains. I will call this small section the X-region in my review below.

The authors conclusions that this X-region represents “a completely new metal-like monoclinic phase at room temperature” (quoted from abstract) rests on two observations in particular:

1. SAED patterns taken of the X-region that indicate that the crystallography is primarily monoclinic (i.e. monoclinic peaks are indeed clearly present in the SAED pattern of this region)
2. s-SNIM images of the nanobeam show that the X-region has reflectivity at ~10 microns that differ from both the equilibrium M1 and R regions. Reflectivity in the X-region is higher than that of the M1 phase, but lower than the R-phase. DFM measurements seem to indicate something similar to the s-SNIM images for the free-carrier density in the three regions.

This reviewer does not find this data sufficient to support the conclusions drawn for two reasons:

1. The authors have not ruled out that the X-region is simply mixed phase M1/R. The X-region could just be nano-stripes (M1/R) or otherwise nanodomain M1 and R regions. Such a mixed phase region could easily give images of enhanced IR reflectivity/scattering in s-SNIM and SAED patterns showing monoclinic peaks. Note: the resolution of the s-SNIM images/instrument is not given in the manuscript, and no quantitative analysis of the SAED patterns is provided.
2. If point 1 can be addressed, current data is still insufficient to prove that AA treatment is responsible for stabilizing the monoclinic metal phase. The authors must describe what kind of sample preparation/processing was required to turn the samples shown in Fig. 1b) into the ‘catlevered’ samples shown in Fig. 2 a)-c)? Vanadium dioxide is very sensitive to strain and it is well known that the MIT depends sensitively on the substrate. The authors should present the results of a proper control experiment and describe the processing steps required to prepare the samples for SAED experiments in order to support the claim that AA treatment (rather than sample processing) is primarily responsible for any of the observations presented.

Additional Comments:

- Temper the language. This is not a 'completely new' phase if something similar was previously observed.

-The authors seem to be using charge density, electron density and charge carrier density interchangeably in the manuscript. These terms are absolutely not interchangeable. In almost all instances I believe that the authors mean either “free carrier density” or “charge carrier density”

-The relationship/registration (scale bars and dimensions) of the images in Fig. 2 should be made more clear.

- The authors don't define what they mean by treatment time? In the treatment procedure described in the Methods section, what is being referred to as the “treatment time” as it appears in Fig. 1b?

- The VASP calculations described in the manuscript and the supplementary material cannot capture the physics of a doped, unconventional Mott insulator (i.e. VO₂). Thus, this reviewer does not think that the results of these calculations provide any support at all for the conclusions being drawn. In addition, there is a great deal of speculation about what must be the nature of the electronic phase transition associated with the monoclinic metallic phase . This discussion is – by and large – borrowed from other articles in the field; the current observations provide neither support for nor a test of any of these notions. In a letter style paper there is always the problem of mixing results (data and evidence) and discussion/speculation, but this reviewer thinks that the current manuscript suffers from this problem in a rather extreme way. The authors should do a better job of separating what that data says unambiguously from what they speculate might be the case at the level of electronic structure.

In summary, this article is on an extremely interesting topic, but the conclusions are not sufficiently supported by the data presented and therefore the manuscript cannot be published in its current form. In addition, there are also a number of other deficiencies with the current form of the manuscript (described above) that should also be addressed before publication.

Reviewer #2 (Remarks to the Author):

This manuscript reports on a new phase in VO₂ MIT induced by surface adsorption of ascorbic acid, which is a reducing agent in general. The novelty of this work are: (1) the revealing of the metal-like metastable phase M1*, which has not been observed in experiments previously. This intermediate state is very important in identifying the nature of the VO₂ metal-to-insulator transition. (2) the method of using surface chemistry to stabilize/capture metastable phase in inorganic materials. Similar concept, i.e. using a chemical reagent to stabilize and thus capture intermediate states in chemical reaction/transformation, has been widely used in protein chemistry/chemical biology. But such idea has been rarely mentioned in phase transition in materials. This manuscript presents a very good example. This reviewer believes the methodology can be widely applied to other metal oxides systems and may lead to highly interesting things in the future.

Based on these observations, this reviewer recommends for the publication of the manuscript in Nature Communication; however, a couple of concerns need to be addressed before it can be accepted: (1) how general is the surface adsorption method for the VO₂ nanobelt material? Have the authors tried other molecules? Do the results show any correlation between the standard reduction potential of the adsorbate and the amount of injected carrier density into VO₂? (2) why was the V_g in DFM measurements selected to be -4V? Do the authors have the DFM data on the VO₂ nanobelt after heating treatment?

Finally, the writing of the manuscript needs improvement. Professional editing services seem necessary to get rid of the numerous incorrect usage of language.

Note

Manuscript ID: NCOMMS-16-28043-T

Title: *Imaging metal-like monoclinic phase stabilized by surface coordination effect in VO₂ nanobeam*

Point-to-point Response to Reviewer #1:

Overall comments from Reviewer #1:

In this manuscript the authors present a variety of measurements on vanadium dioxide (VO₂) nanobeams that were subject to a treatment procedure involving L-Ascorbic acid (AA). The authors interpret these results as suggesting that the AA treatment stabilizes a monoclinic metallic phase of VO₂ at room temperature (similar to that previously observed in time-resolved studies of photoexcited VO₂, Ref.20 in the current manuscript) along a small section of the treated nanobeam located between insulating M1 and metallic R domains. I will call this small section the X-region in my review below. In summary, this article is on an extremely interesting topic, but the conclusions are not sufficiently supported by the data presented and therefore the manuscript cannot be published in its current form. In addition, there are also a number of other deficiencies with the current form of the manuscript that should also be addressed before publication.

Authors' response:

We thank the reviewer's constructive suggestions and positive comments on our work. We have made changes according to the constructive suggestions. Detailed responses are provided in the following.

Comment 1: The authors have not ruled out that the X-region is simply mixed phase M1/R. The X-region could just be nano-stripes (M1/R) or otherwise nanodomain M1 and R regions. Such a mixed phase region could easily give images of enhanced IR reflectivity/scattering in s-SNIM and SAED patterns showing monoclinic peaks.

Note: the resolution of the s-SNIM images/instrument is not given in the manuscript, and no quantitative analysis of the SAED patterns is provided.

Authors' response 1: Thanks for the reviewer's kind suggestions. To rule out the possibility of a mixed M1 and R phase in the X-region observed in our experiment, we give the further detailed responses from the following three aspects: **(i)** From the images of s-SNIM and DFM, we can see that the signal of the X-region were uniform, indicating single homogeneous phase in this region. **(ii)** The irreversible transition of the X-region after heating treatment illustrated that the X-region is not a mixed M1 and R phase. **(iii)** The added Raman spectra further verify that the X-region is a homogeneous metastable phase not a mixed M1/R phase.

The detailed responses are provided in the following:

(i) Scattering scanning near-field infrared microscope (s-SNIM) is an ultrahigh-resolution optical microscopy system, relying on light scattering by the probing tip of an AFM to obtained nanoscale near-field images. In our work, the near-field images of the AA-treated VO₂ nanobeam were performed on NeaSNOM (Neaspec GmbH Co.) with the **resolution of about 10 nm**. In addition, the dielectric force microscopy (DFM) is also a contactless imaging technique with **nanometer-scaled spatial resolution**. From the images of s-SNIM and DFM in our manuscript, it can be clearly seen that the signal of the X-region is uniform, suggesting a single homogenous phase in this region.

(ii) If the X-region is a mixed M1/R phase, the increased rutile domain would return to the mixed M1/R phase when cooling to room temperature again, because the transition between R and M1 phase is reversible. From the optical images, SAED and s-SNIM images after the heating treatment, we find that the increased rutile domain transformed from the X-region can't turn into the initial state and keep rutile phase after cooling to room temperature. That is to say, **the transition of the X-region through the heating treatment is irreversible, demonstrating the X-region is not a mixed M1/R phase but a homogenous metastable phase**.

(iii) Raman spectrum, as it is sensitive to the nanomaterials' structure, has been a powerful tool to identify different phases of materials. The Raman spectra of VO₂

(M1) and VO₂ (R) are completely different, which we can determine the phases of VO₂ easily through the characteristic peak of their Raman spectra. According to the reviewer's concern, we further provide the Raman spectra of the suspended metal-like monoclinic domain (X-region) before and after heating treatment. As shown in **Figure N1**, we can see that the measured region exhibit the characteristic peaks of VO₂ (M1) phase before heating treatment. After heating the nanobeam up to 350K and cooling to room temperature again, the Raman spectra of the same region match well with the VO₂ (R) and no peaks of M1 phase can be found [Nano lett. 2009, 9, 4527]. That is to say, **no M1 phase of VO₂ exist in the X-region after heating treatment**. Therefore, we can rule out that the X-region is a mixed M1/R phase.

On the reviewer's concern of the quantitative analysis of the SAED patterns, we have indexed the SAED patterns of this metastable X-region before and after heating treatment in Fig. 2 of the revised manuscript, which correspond to the monoclinic and rutile structure, respectively.

Accordingly, to further confirm the observed metal-like monoclinic phase in the AA-treated VO₂ nanobeams, we have added the corresponding statements in the revised manuscript. We have given the resolution of the s-SNIM and indexed the SAED patterns in the revised manuscript. Also, the Raman spectra of the X-region before and after heating treatment have been added in the revised supplementary information.

Figure N1. a and b, Room-temperature Raman spectra of the metastable region

monitored before and after heating treatment, respectively.

Comment 2: If point 1 can be addressed, current data is still insufficient to prove that AA treatment is responsible for stabilizing the monoclinic metal phase. The authors must describe what kind of sample preparation/processing was required to turn the samples shown in Fig. 1b) into the ‘catilevered’ samples shown in Fig. 2 a)-c)? Vanadium dioxide is very sensitive to strain and it is well known that the MIT depends sensitively on the substrate. The authors should present the results of a proper control experiment and describe the processing steps required to prepare the samples for SAED experiments in order to support the claim that AA treatment (rather than sample processing) is primarily responsible for any of the observations presented.

Authors’ response 2: We appreciate the reviewer’s kind suggestions. According to the reviewer’s constructive suggestion, we have performed the control experiments and added detailed descriptions of the experimental processing in the revised manuscript. The detailed explanations have been given point-by-point as follows.

(1) About the reviewer’s concern about “The authors must describe what kind of sample preparation/processing was required to turn the samples shown in Fig. 1b) into the ‘catilevered’ samples shown in Fig. 2 a)-c)?”

In our case, the AA-treated VO₂ nanobeams were transferred onto the edge of silicon substrate using silver needle under Olympus optical microscope. In a typical procedure, the AA-treated VO₂ nanobeams were obtained by treated in L-ascorbic acid (AA) aqueous solution at 80 °C with nitrogen protecting as shown in Fig. 1b. As is known, the MIT of VO₂ is sensitive to substrate strain. Thus, before heating treatment, we transfer the AA-treated VO₂ nanobeams onto the edge of silicon substrate and make one end of the nanobeams protruding over the edge of the substrate to eliminate the effect of substrate strain as shown in Fig. 2 a)-c). The silver needle was used to transfer the AA-treated VO₂ nanobeams. To better understand the transfer processing, we provided the schematic illustration and optical image as

shown in **Figure N2**. Firstly, we slowly approach and touch the AA-treated VO₂ nanobeam using silver needle. Then AA-treated VO₂ nanobeam could be adsorbed on the silver needle. Subsequently, moving the silver needle slowly to the edge of a new SiO₂/Si substrate and put down the nanobeam on the edge of the substrate. Finally, slight adjustment is required to make one end of the AA-treated VO₂ nanobeam suspend on the edge of substrate. Of note, multiple attempts are needed for the successful transfer.

According to reviewer's suggestion, we have added the necessary description of the sample transferring process in the revised manuscript Methods section as "The AA-treated VO₂ nanobeam was transferred onto the edge of silicon substrate using silver needle under Olympus optical microscope. In a typical procedure, the silver needle slowly approach and touch the AA-treated VO₂ nanobeam on the SiO₂/Si substrate. Then the AA-treated VO₂ nanobeam could be adsorbed on tip of the silver needle. Subsequently, moving the silver needle slowly to the edge of another SiO₂/Si substrate and put down the nanobeam on the edge of the substrate. Finally, slight adjustment made one end of the AA-treated VO₂ nanobeam suspend on the edge of substrate."

Figure N2. Schematic illustration of the sample transferring process. The inset showed the optical image of the AA-treated VO₂ nanobeam adsorbed on the silver

needle.

(2) About the reviewer's concern about "The authors should present the results of a proper control experiment"

In order to further confirm that the stabilization of rutile metal and metastable monoclinic metal phases is due to the AA molecules treatment, we have provided the control experiments. We treated the VO₂ nanobeams in the pure water without adding AA and kept other conditions unchanged. **Figure N3** shows the optical image and Raman spectra of VO₂ nanobeam treated in pure water. We can see that the VO₂ nanobeam exhibited a homogeneous bright reflection of monoclinic insulating phase and no dark domains emerged at the two ends of the nanobeam. The Raman spectra further demonstrate the VO₂ nanobeam kept monoclinic phase at room temperature after treatment in pure water. Thus, these results of the control experiments indicate the emergency and stabilization of rutile metal phase and monoclinic metal phase at room temperature is attributed to the AA treatment in our case.

According to reviewer's suggestion, the control experiments have been added in the revised supplementary information.

Figure N3. Optical image and Raman spectra of the VO₂ nanobeam treated in pure water. The peak labeled by "*" in Raman spectra belongs to the silicon substrate.

(3) About the reviewer's concern about "describe the processing steps required to prepare the samples for SAED experiments in order to support the claim that AA treatment (rather than sample processing) is primarily responsible for any of the

observations presented.”

In our work, to perform the SAED measurements, we transferred the AA-treated VO₂ nanobeams to the copper grid using silver needle as described above. Before transfer, we initially took the optical image of the AA-treated VO₂ nanobeam on the SiO₂/Si substrate and recorded the length of the dark R domain. Then we transferred the nanobeam onto the copper grid for the SAED measurements as shown in **Figure N4**. In the typical SAED measurements, we first measured the length from the end of nanobeam and determined the boundary of dark and bright domains according to the optical image. Then we performed the SAED on the metastable region adjacent to the boundary and marked this position for following experiment as shown in the inset of Fig. 2a in the original manuscript. Subsequently, the copper grid was taken out from the SAED instrument and heated on a hot plate up to 350 K. After cooling to room temperature, the copper grid was put back into the SAED instrument and found the early marked region on the AA-treated nanobeam for SAED. And, the SAED was performed at the same region as shown in the inset of Fig. 2c in the original manuscript. The SAED results illustrated that this small region adjacent to metallic R domain is a metastable phase and retained monoclinic structure, which is consistent with the optical change before and after heating.

According to reviewer’s suggestion, the processing steps for SAED measurements have been added in Methods section of revised manuscript as “The optical image of AA-treated VO₂ nanobeam on the SiO₂/Si substrate was taken and recorded the length of the dark R domains along the nanobeam. Then, the AA-treated VO₂ nanobeam was transferred onto the copper grid using silver needle. During the SAED measurements, the boundary of dark and bright domains was first determined according to the optical image. Then the SAED for the metastable region adjacent to the boundary was performed and marked the corresponding position. Subsequently, the copper grid was taken out and heated on a hot plate up to 350 K. After cooling to room temperature, the copper grid was put back and found the same region on the nanobeam as before heating treatment for SAED. Finally, the SAED pattern on the same region was obtained.”

Figure N4. Optical image of the AA-treated VO₂ nanobeam on the copper grid for SAED measurements.

In conclusion, all the results demonstrate that the effect of substrate strain is carefully ruled out and AA treatment is responsible for stabilizing the monoclinic metal phase. According to the reviewer's valuable suggestions, we have added the experimental details and control experiments in the revised manuscript and supplementary information.

Comment 3: Temper the language. This is not a 'completely new' phase if something similar was previously observed.

Authors' response 3: We agree with referee's opinion that the description of "completely-new" is not appropriate in our original manuscript. In order to avoid the misunderstanding, we have updated the corresponding description into "unusual" in the revised manuscript.

Comment 4: The authors seem to be using charge density, electron density and charge carrier density interchangeably in the manuscript. These terms are absolutely not interchangeable. In almost all instances I believe that the authors mean either "free carrier density" or "charge carrier density".

Authors' response 4: Thanks for the reviewer's valuable comments. According to the reviewer's kind suggestion, we have corrected the corresponding terms into "charge carrier density" in the revised manuscript.

Comment 5: The relationship/registration (scale bars and dimensions) of the images in Fig. 2 should be made more clear.

Authors' response 5: We appreciate reviewer's kind suggestions. According to reviewer's reminding, we have adjusted the images and added the corresponding scale bars in Fig.2 in the revised manuscript.

Comment 6: The authors don't define what they mean by treatment time? In the treatment procedure described in the Methods section, what is being referred to as the "treatment time" as it appears in Fig. 1b?

Authors' response 6: Thanks for the reviewer's kind suggestions. In fact, the "treatment time" in our original manuscript referred to the reflux reaction time for the VO₂ nanobeams in the AA aqueous solution at 80 °C with nitrogen protecting. The reaction time for AA-treated VO₂ nanobeams in Fig. 1b from left to right in turn is 0, 1, 3, 5 and 8h. In order to avoid the misunderstanding, we have defined the treatment time in the Methods section of the revised manuscript.

Comment 7: The VASP calculations described in the manuscript and the supplementary material cannot capture the physics of a doped, unconventional Mott insulator (i.e. VO₂). Thus, this reviewer does not think that the results of these calculations provide any support at all for the conclusions being drawn.

Authors' response 7: We appreciate reviewer's kind comments. According to reviewer's concern, we realized that the strongly correlated system is rather complicated, especially, in a doped, unconventional system. Thus, we agree with the reviewer's opinion that the VASP calculations might not be suitable to our work. Nevertheless, in our case, the stabilization of the metal-like monoclinic phase can be well understood from previous reports and our careful characterizations. There is a general consensus that the occupancy of t_{2g} orbitals determined the electronic phases of VO₂ [*Phys. Rev. Lett.* 2005, 94, 026404; *Phys. Rev. Lett.* 2012, 108, 256402; *Science* 2014, 346, 445]. In our case, the observed metal-like monoclinic phase in the

AA-treated VO₂ nanobeam is a new intermediate phase. Thus, we deem that the occupancy of t_{2g} orbitals is responsible to the stabilization of metal-like monoclinic VO₂ phase. Notably, in our present work, the electron-donating AA molecules and the DFM measurements indicate the electron injection into VO₂ nanobeams during the AA treatment. The injected electrons would alter the charge carrier density of the three t_{2g} orbitals, inducing charge carrier density reorganization. Thus, combining our experimental data and previous reports, we declared that the reorganization of charge carrier density of t_{2g} orbitals induced the metal-like monoclinic phase in our AA-treated VO₂ nanobeams. Our finds contribute to an in-depth understanding of VO₂ MIT, and will be of interest to the regulation of low-dimensional strongly-correlated solids.

Accordingly, the discussion from the previous literatures and our characterizations provide a reasonable explanation of the stabilization of metal-like monoclinic VO₂ phase and thus we removed the VASP calculations in the revised manuscript.

Comment 8: there is a great deal of speculation about what must be the nature of the electronic phase transition associated with the monoclinic metallic phase. This discussion is – by and large – borrowed from other articles in the field; the current observations provide neither support for nor a test of any of these notions. In a letter style paper there is always the problem of mixing results (data and evidence) and discussion/speculation, but this reviewer thinks that the current manuscript suffers from this problem in a rather extreme way. The authors should do a better job of separating what that data says unambiguously from what they speculate might be the case at the level of electronic structure.

Authors' response 8: Thanks for the reviewer's helpful comments. In order to give clear information and statements of our work, we have carefully adjusted the format of manuscript and separated the data and discussion sections according to the reviewer's suggestions and editorial policies of Nature Communications. We believe that the revised manuscript is now more suitable for publication.

Point-to-point Response to Reviewer #2:

Overall comments from Reviewer #2:

This manuscript reports on a new phase in VO₂ MIT induced by surface adsorption of ascorbic acid, which is a reducing agent in general. The novelty of this work are: (1) the revealing of the metal-like metastable phase M1*, which has not been observed in experiments previously. This intermediate state is very important in identifying the nature of the VO₂ metal-to-insulator transition. (2) the method of using surface chemistry to stabilize/capture metastable phase in inorganic materials. Similar concept, i.e. using a chemical reagent to stabilize and thus capture intermediate states in chemical reaction/transformation, has been widely used in protein chemistry/chemical biology. But such idea has been rarely mentioned in phase transition in materials. This manuscript presents a very good example. This reviewer believes the methodology can be widely applied to other metal oxides systems and may lead to highly interesting things in the future. Based on these observations, this reviewer recommends for the publication of the manuscript in Nature Communication; however, a couple of concerns need to be addressed before it can be accepted.

Authors' response:

We are grateful to the reviewer's comments on the significance of our work. We have made changes according to the constructive suggestions. Detailed responses are provided in the following.

Comment 1: how general is the surface adsorption method for the VO₂ nanobelt material? Have the authors tried other molecules? Do the results show any correlation between the standard reduction potential of the adsorbate and the amount of injected carrier density into VO₂?

Authors' response 1: According to reviewer's suggestion, we have tried some other electron-donating molecules, such as oxalic acid, ethylene glycol, glucose and NaBH₄, to regulate the metal-insulator transition (MIT) of VO₂ nanobeams. However, the MIT modulation of VO₂ nanobeams has not been observed. The optical images and corresponding Raman spectra of the VO₂ nanobeams treated in oxalic acid, ethylene

glycol, glucose and NaBH₄ solution have been illustrated in **Figure N5**. It can be seen that the VO₂ nanobeams exhibited homogenous bright reflection with no color change and the Raman spectra indicated that the VO₂ nanobeams maintained the monoclinic phase after the treatments. That is to say, these molecules treatments could not trigger the phase evolution of VO₂ nanobeams as it appears in L-ascorbic acid treatment.

In our L-ascorbic acid treatment, the phase evolution and stabilization of metal-like monoclinic phase in VO₂ is induced by the interfacial charge transfer. In this regard, we deem that the interaction between L-ascorbic acid and VO₂ nanobeam is a complicated process and effective regulation of VO₂ MIT using chemical molecules needs two critical factors. First, the treated molecules could coordinately bond on the surface of VO₂ nanobeams to form a stable complex. Second, the suitable match of energy level between the chelated molecules and VO₂ is indispensable to induce the charge transfer. We believe that the utilization of suitable molecules holds great promise for engineering properties of low-dimensional correlated electron solids.

According to the reviewer's suggestion, we have added the above contrast experiments treated with other molecules and corresponding statements in the revised supplementary information.

Figure N5. a-d, Optical images and Raman spectra of the VO₂ nanobeams treated in oxalic acid, ethylene glycol, glucose and NaBH₄ solution. Note: the reducing ability of oxalic acid, ethylene glycol and glucose is weaker than L-ascorbic acid, while

reducing ability of NaBH_4 is stronger than L-ascorbic acid. The peak labeled by “*” in Raman spectra belongs to the silicon substrate.

Comment 2: why was the V_g in DFM measurements selected to be -4V? Do the authors have the DFM data on the VO_2 nanobelt after heating treatment?

Authors' response 2: We appreciate the reviewer's kind comments. According to the reviewer's concern, we have provided the corresponding statements and added the quantitative DFM signal at different V_g in the revised manuscript. The detailed explanations have been given point-by-point as follows.

(1) About the reviewer's concern about “why was the V_g in DFM measurements selected to be -4V?”

We thank reviewer for comment on DFM measurement. In fact, we have performed DFM measurements at multiple different gate voltages (V_g) from -4V to +4V. Here, gate voltage $V_g = -4\text{V}$ is selected as a typical imaging parameter of DFM measurement presented in our original manuscript. The quantitative DFM signal at different V_g has been shown in **Figure N6**, it is shown that there are three different DFM response regions in all V_g , which confirms that the metal-like monoclinic intermediate phase has higher carrier density than mid phase, and lower carrier density than end rutile phase. Thus, the results of DFM measurements are consistent with that of the s-SNIM and clearly demonstrate three different charge carrier density in our AA-treated VO_2 nanobeams.

According to the reviewer's concern, we have added the corresponding statements in the revised manuscript and added the quantitative DFM signal at different V_g in the revised supplementary information.

Figure N6. DFM signal versus V_g plot from -4V to +4V for end metallic rutile, metal-like monoclinic and mid insulating monoclinic domains.

(2) About the reviewer’s concern about “Do the authors have the DFM data on the VO₂ nanobelt after heating treatment?”

According to the reviewer’s concern, we provided the DFM response of the AA-treated VO₂ nanobeam after heating treatment. DFM image at $V_g = -4V$ on the AA-treated VO₂ nanobeam through heating treatment is shown in **Figure N7**. It is clearly found that the dielectric response exhibits two different DFM signal regions, indicating two different charge density regions. Thus, this DFM data is consistent with the results of s-SNIM, optical contrast and SAED in the manuscript, which demonstrates that metal-like monoclinic VO₂ converted to metallic rutile phase after heating treatment.

Figure N7. DFM image of the AA-treated VO₂ nanobeam after heating treatment at

$$V_g = -4V.$$

Comment 3: the writing of the manuscript needs improvement. Professional editing services seems necessary to get rid of the numerous incorrect usage of language.

Authors' response 3: Thanks very much for the reviewer's kind suggestions. The grammar and spelling mistakes, as well as the expression have been carefully edited in the revised manuscript. We believe that the revised manuscript is now more qualified for publication.

List of changes made to the manuscript

(1) According to the comments by Reviewer 1, we have added the resolution of s-SNIM and indexed the SAED in the revised manuscript. We have also added the corresponding analysis and Raman spectra before and after heating treatment to rule out the observed metastable region is a mix M1/R phases in the revised manuscript and supplementary information.

(2) According to the suggestions by Reviewer 1, we have added the detailed description of the sample transferring process in the revised manuscript Methods section. As suggested by the reviewer, we have provided the control experiment in the revised supplementary information to verify that the stabilization of metal-like monoclinic phase at room temperature is attributed to the AA treatment. We have also added the detailed description of the processing steps for SAED measurements in the revised manuscript Methods section.

(3) According to the suggestions by Reviewer 1, we have checked the manuscript and updated the inappropriate terms in the revised manuscript. Such as, changed the “completely-new” to “unusual”; corrected the “charge density” and “electron density” to “charge carrier density”; defined the “treatment time” as “reflux reaction time” and so on.

(4) According to the suggestions by Reviewer 1, we have adjusted the images and added the corresponding scale bars in Fig.2 in the revised manuscript.

(5) According to the comments by Reviewer 1, we have removed the VASP calculations in the revised manuscript.

(6) According to the suggestions by Reviewer 1, we have adjusted the format of manuscript and separated the data and discussion sections in the revised manuscript.

(7) According to the comments by Reviewer 2, we have added the contrast experiments treated with other molecules and corresponding statements in the revised manuscript and supplementary information.

(8) According to the comments by Reviewer 2, we have added the corresponding statements why selected $V_g = -4V$ in the DFM measurements in the revised manuscript and added the quantitative DFM signal at different V_g in the revised supplementary information.

(9) According to the suggestions by Reviewer 2, we have revised the grammar and spelling mistakes in the revised manuscript and supporting information.

REVIEWERS' COMMENTS:

Reviewer #1 (Remarks to the Author):

My primary criticism of the manuscript was related to the interpretation of the data on the X-region. In my opinion, the data were insufficient to rule out that this region was mixed phase M1/R rather than a more exotic monoclinic metallic phase/state of VO₂.

In response to this criticism the authors note:

To rule out the possibility of a mixed M1 and R phase in the X-region observed in our experiment, we give the further detailed responses from the following three aspects: (i) From the images of s-SNIM and DFM, we can see that the signal of the X-region were uniform, indicating single homogeneous phase in this region. (ii) The irreversible transition of the X-region after heating treatment illustrated that the X-region is not a mixed M1 and R phase. (iii) The added Raman spectra further verify that the X-region is a homogeneous metastable phase not a mixed M1/R phase.

Of these three points, only the first is convincing. It does seem that the resolution of the s-SNIM and DFM imaging methods used should be sufficient to image domains if they were present (unless these domains fluctuate?), even if the images presented are rather noisy..... especially the DFM images (Fig. 3), which have a profoundly striped appearance that are unexplained in the manuscript? The observations referenced in the second and third points in no way rule out the X-region being of mixed phase following AA treatment. On balance, however, I am satisfied that the authors have made an effort to address this point in the revision.

Note: my point regarding a 'quantitative analysis of the SAED patterns' was not to encourage the authors to index the diffraction patterns, but instead to stimulate an analysis of the peak intensities to see if the pattern taken of the X-region is best understood as mixed M1/R or pure M1 phase.

Regardless, given all the data it does seem most likely that the X-region is a homogeneous metastable phase. The text of the article should, however, reflect the tentative nature of that conclusion rather than being so definitive on this point (as it is now written).

Lastly, it seems necessary to add a discussion – even if this discussion is speculative – of why the monoclinic metallic phase in the X-region should be limited to such a small section of the nanobeam between the R-phase region (that 'grow in' from the ends with treatment time) and the remaining M1 phase material. Do the authors have a model in mind for why the stabilized R-phase region grows in from the ends of the nanobeam with treatment time? How are we to understand this phenomena from the perspective of the adsorption of AA and electron transfer at the surface of the nanobeam as described? Currently the manuscript is silent on this matter. This should be addressed before publication.

Reviewer #2 (Remarks to the Author):

The questions and concerns I raised in the previous review have been addressed. I am satisfied with the revised manuscript and recommend it for publication in Nature Communications.

Note

Manuscript ID: NCOMMS-16-28043A

Title: *Imaging metal-like monoclinic phase stabilized by surface coordination effect in vanadium dioxide nanobeam*

Point-to-point Response to Reviewer #1:

Authors' response:

We thank the reviewer's helpful suggestions and positive comments on our work and previous revision. We have made corresponding discussion and changes to address the remaining concerns. Detailed responses are provided in the following.

Comment 1: It does seem that the resolution of the s-SNIM and DFM imaging methods used should be sufficient to image domains if they were present (unless these domains fluctuate?), even if the images presented are rather noisy..... especially the DFM images (Fig. 3), which have a profoundly striped appearance that are unexplained in the manuscript?

Authors' response 1: We appreciate the reviewer's reminding. In fact, the striped appearance in the DFM image is mainly caused by the following two factors: (1) the first reason originated from the slight fluctuation of DFM signal and the accompanying noise of the DFM measurement system. (2) The second reason came from a bit thickness of our AA-treated VO₂ nanobeam. We have made our efforts to adjust flatten process of DFM image carefully to reduce the striped appearance in the DFM measurements. Whereas, current resolution in our work is sufficient to recognize the corresponding domains and three different types of DFM signal strength can be clearly observed in the DFM image.

According to reviewer's suggestion, we have added the corresponding explanation in the revised manuscript.

Comment 2: my point regarding a ‘quantitative analysis of the SAED patterns’ was not to encourage the authors to index the diffraction patterns, but instead to stimulate an analysis of the peak intensities to see if the pattern taken of the X-region is best understood as mixed M1/R or pure M1 phase.

Authors’ response 2: According to the reviewer’s suggestion, we have provided the analysis of peak intensities of the SAED patterns obtained in the X-region and the pristine M1 phase. Figure N1a and b shows the SAED patterns obtained in the X-region and pristine M1 phase, respectively. We can see that the X-region exhibits the same SAED pattern as the pristine M1 phase. Furthermore, the peak intensities were obtained from the ED spots marked by the red and green dashed lines. It can be seen that the ratio of peak intensities of different SAED spots in X-region bear resemblance to that of the pristine M1 phase. These results indicate that the X-region exhibits homogeneous M1 structure.

Figure N1. Analysis of the peak intensities of the SAED patterns obtained in the

X-region and the pristine M1 phase.

Comment 3: Regardless, given all the data it does seem most likely that the X-region is a homogeneous metastable phase. The text of the article should, however, reflect the tentative nature of that conclusion rather than being so definitive on this point.

Authors' response 3: Thanks for the reviewer's positive review of our opinion. Besides the s-SNIM and DFM images, however, we deem that the irreversible transition of the X-region after heating treatment and the added Raman spectra gave further support for metastable monoclinic metallic phase in our case. In order to better understand our observations, we provide further discussions as following.

As is known, VO₂ undergoes a reversible first-order metal-insulator transition (MIT) near room temperature of ~ 340 K, accompanying with a lattice change from rutile (R) to monoclinic (M1). In our AA-treated VO₂ nanobeam, through a heating treatment as shown in Figure 2a-c in the original manuscript, we can see that the X-region became rutile phase after cooling to room temperature again. From the SAED pattern and Raman spectra of this increased R domain transforming from the X-region, we can identify that no component of M1 phase was found because of the obvious difference of SAED pattern and Raman spectra between M1 phase and R phase. That is to say, **this X-region underwent an irreversible transition and became a homogeneous R phase after the heating treatment. However, if the X-region is a mixed M1/R phase, it will become a mixed M1/R phase again after cooling to room temperature.** The SAED pattern and Raman spectra will show the characteristic pattern and peaks of M1 phase. This is contradictory with our observations.

In a word, the irreversible transition indicates that the X-region is not a mixed M1/R phase, but a homogeneous metastable phase.

Comment 4: it seems necessary to add a discussion – even if this discussion is speculative – of why the monoclinic metallic phase in the X-region should be limited to such a small section of the nanobeam between the R-phase region (that 'grow in' from the ends with treatment time) and the remaining M1 phase material.

Authors' response 4: We appreciate reviewer's kind suggestions. For the small section of metastable monoclinic metallic phase, we can understand it from the following two aspects:

(i) Stabilization of monoclinic metallic phase limited in a narrow range of the electron density. In our case, the chelated AA molecules transfer electrons into VO₂ nanobeams. The optical images suggest that the electrons injection is along the V-V atomic chain direction from the ends of VO₂ nanobeam to the middle. Thus, the electron density almost certainly varies smoothly as a function of position away from the ends of nanobeam. That is to say, the electron density near the ends of the AA-treated VO₂ nanobeam is maximum and gradually reduces towards the middle. From the s-SNIM and DFM images, we can see that the monoclinic metallic phase exists in a small part of the AA-treated VO₂ nanobeam between the stabilized R phase domain and initial M1 phase domain. This suggests that there may be a specifically small range of the electron density necessary to stabilize the monoclinic metallic phase. When the electron density exceeds this range, the R phase would be stabilized; while the electron density is lower than this density range, the VO₂ keeps monoclinic insulating phase.

(ii) Metastable monoclinic metallic phase is difficult to exist in a large region. As is known, the metastable materials are very sensitive to external perturbation and are thermodynamic instability. Thus, it is difficult to form a large-area metastable domain.

According to reviewer's suggestion, we have added the corresponding discussion of the small VO₂ monoclinic metallic domain in the revised manuscript and revised supplementary information as "In our case, the chelated AA molecules transferred electrons into VO₂ nanobeams along the V-V atomic chain direction. The electron density varies as a function of position away from the ends of AA-treated VO₂ nanobeam. The monoclinic metallic domain exists in a small part of the nanobeam between the stabilized R phase domain and initial M1 phase domain. This suggests that there may be a specifically small range of electron density necessary to stabilize metal-like monoclinic phase."

Comment 5: Do the authors have a model in mind for why the stabilized R-phase region grows in from the ends of the nanobeam with treatment time? How are we to understand this phenomena from the perspective of the adsorption of AA and electron transfer at the surface of the nanobeam as described?

Authors' response 5: Thanks for the reviewer's suggestions. From the optical images, we can see that the stabilized R phase expand from the ends of the VO₂ nanobeam to the middle with treatment time. Considering the crystal structure and 3d orbital overlap of VO₂, we deem that this phenomenon can be understood from the fact that the injected electrons transfer along the V-V atomic chains from the ends to the middle. VO₂ is a typical strongly correlated oxide, which consists of infinite V-V atomic chains along the rutile [001]_R direction (monoclinic [100]_{M1}). Specially, in VO₂ nanobeams, the V-V atomic chains are elongated along the nanobeam growth direction.

As shown in Figure N2, VO₆ octahedra share edges to form one-dimensional chain with the V3d orbitals pointing to neighbouring vanadium ions. The electronic states of these correlated V-V atomic chains determined the properties of VO₂. In our case, the AA molecules exhibit strong electron-donating and coordination ability. The AA molecules bind with the V atoms on the treated VO₂ nanobeams, which have been verified by the ATR-FTIR spectra. Observing from the optical and DFM images, the electrons injected into VO₂ nanobeam from the ends to the middle. Therefore, the AA molecules binding on the end surface of VO₂ nanobeams dominantly contributed to the electrons injection. Due to the energy matching and overlap of the V3d orbitals along the V-V atomic chains, electrons can transfer from AA to VO₂ nanobeams along the chain direction (Figure N2). When the electron density reach a critical value, R-phase domain could be stabilized. With the treatment time, the R-phase domains expand towards the middle.

According to reviewer's suggestion, we have added the corresponding discussion in the revised manuscript and revised supplementary information as "From the optical and DFM images, it can be understood that the AA molecules binding on the end surface of VO₂ nanobeams contributed electrons injection to VO₂ nanobeams. Due to

the energy matching and overlap of the $V3d$ orbitals along the V-V atomic chains, electrons can transfer from AA to VO_2 nanobeams along the chain direction. Thus, the R-phase domain grows in from the ends of the nanobeam with treatment time.”

Figure N2. Schematic model of the electrons transfer from the ends to the middle of VO_2 nanobeam.

Response to Reviewer #2:

Comment from Reviewer #2:

The questions and concerns I raised in the previous review have been addressed. I am satisfied with the revised manuscript and recommend it for publication in *Nature Communications*.

Authors' response:

We thank the Reviewer for the positive comments and recommending our work to be published. We believe that the revised manuscript is now more qualified for publication in the respected journal *Nature Communications*.

List of changes made to the manuscript

(1) According to the comments by Reviewer 1, we have added the corresponding explanation of the striped appearance in the DFM image in the revised manuscript.

(2) According to the suggestions by Reviewer 1, we have added the corresponding discussion of the small VO₂ monoclinic metallic domain in the revised manuscript and revised supplementary information.

(3) According to the suggestions by Reviewer 1, we have added the corresponding discussion that stabilized R-phase region grows in from the ends of the nanobeam in the revised manuscript and revised supplementary information.